# Liquid Crystals Investigation Behavior on Azo-Based Compounds: A Review

**DOI:** 10.3390/polym13203462

**Published:** 2021-10-09

**Authors:** Nurul Asma Razali, Zuhair Jamain

**Affiliations:** Sustainable Materials and Renewable Energy (SMRE) Research Group, Faculty of Science and Natural Resources, Universiti Malaysia Sabah, Jalan UMS, Kota Kinabalu 88400, Sabah, Malaysia; nasmarazali@gmail.com

**Keywords:** azo compound, liquid crystal, structure–property relationship

## Abstract

Liquid crystal is an intermediate phase between the crystalline solid and an isotropic liquid, a very common substance in our daily lives. Two major classes of liquid crystal are lyotropic, where a liquid crystal is dissolved in a specific solvent under a particular concentration and thermotropic, which can be observed under temperature difference. This review aims to understand how a structure of a certain azo compound might influence the liquid crystal properties. A few factors influence the formation of different liquid crystals: the length of the alkyl terminal chain, inter/intra-molecular interaction, presence of spacer, spacer length, polarization effects, odd-even effects, and the presence of an electron-withdrawing group or an electron-donating group. As final observations, we show the compound’s different factors, the other liquid crystal is exhibited, and the structure–property relationship is explained. Liquid crystal technology is an ideal system to be applied to products to maximize their use, especially in the electronic and medical areas.

## 1. Introduction

Azo compounds represent a large branch of the liquid crystal sciences. The azo compound is a compound that contains two or more organic groups, which an azo group separates by –N=N– as its linking unit [1,2]. In order to conserve natural flora and fauna sources, an alternative was taken by producing a synthetic dye using an azo compound. An azo compound is an important group used extensively in the textile industry [3], manufacture of ink [4], and the cosmetic industry [5].

Liquid crystal is a liquid that is not isotropic, has two refractive indexes, and displays interference in a polarized light [6,7]. According to Stegemeyer (1989), it was first discovered by Dr. Friedrich Reinitzer back in 1888 in a German University located in Prague, where he revealed two crucial traits of the cholesteric liquid crystal, which are the change of color with temperature and the temperature range of the cholesteric phase, and the melting point and clearing point [8,9].

It has been proven that liquid crystal can form an intermediate phase, called the mesophase. This phase is situated in the middle of a crystalline solid and an isotropic liquid [10]. Relying on a particular circumstance in which the mesophase becomes apparent, liquid crystal can be characterized into two major classes: lyotropic and thermotropic [11]. It can be suggested that lyotropic shows the liquid crystal state in a certain concentration, as Andrienko (2018) reported that the specification of this liquid crystal is its concentration [12].

However, Brightman (1954) has stated that the thermotropic phase is a phase where a liquid crystal is prepared via a heating process [7]. It can be signified as a liquid crystal formed with temperature change, in agreement with Kusabayashi and Takenaka (1984) who reported that thermotropic liquid crystals give out a few different forms with temperature variation [13]. Enantiotropic is a term to define a thermodynamically stable mesophase that appears on heating and cooling of the molecule, while a thermotropic mesophase that arises only during cooling of a molecule is termed as a monotropic mesophase. On the other hand, a molecule that exhibits liquid crystal under the same influence as a lyotropic (solvent) and a thermotropic (heat) is referred to as an amphotropic [14].

Generally, the systemic nature of the thermotropic liquid crystal is dependable on the molecular shape of the crystal, whether it is a rod-like or a disc-like shape molecule [15]. The thermotropic liquid crystal with disc-like molecules are a discotic mesogen, while a thermotropic liquid crystal with rod-like molecules is a calamitic mesogen, which can be sub-divided into nematic, cholesteric and smectic phase.

A nematic phase is the simplest liquid crystal where the molecules are positioned in a long axis across the same preferred direction [12]. This means that the rod-like molecules are arranged in a plane parallelly. In a thermotropic liquid crystal, the positional order of the molecules can be destructed if the molecules are treated with heat at a specific temperature but not the orientational order [15]. As a molecule of a nematic mesophase is aligned parallelly along its axes, this mesophase exhibits anisotropic physical properties. Typically, a nematic phase shows schlieren, marble, and pseudoisotropic textures in accordance with the structure of its surface.

Cholesteric mesophase is also known as a chiral nematic phase (N*) as it is formed via doping of a nematic liquid crystal or when the molecules of a system are chiral [16]. The director of a cholesteric mesogen tends to form a helix with pitch due to the alignment of an adjacent molecule at a trivial angle of one another. Basically, a pitch is a distance taken by the director to make one full turn in the helix. This helical structure of N* can reflect light with a wavelength uniform to the pitch length. A cholesteric mesogen commonly exhibit a schlieren textures.

Different from a nematic phase, a smectic phase is aligned in layers disclosing an association between its position along with the orientational order [12]. A different smectic phase is formed as the molecular order of the smectic phase change, which is Smectic A (SmA), Smectic B (SmB), Smectic C (SmC), Smectic F (SmF), and Smectic I (SmI). These smectic are then sub-divided into two categories depending on whether the molecules are tilted to the layer normal or not. A part of SmA and SmB, other smectic are tilted phase.

Among all other smectic, SmA or SmC are often observed. These mesogen is formed when each molecule does not have a long-range positional order [15]. The molecules in a smectic A phase are arranged in layers so that the long axis is perpendicularly aligned to the plane. Most compound that exhibits these kinds of structure is a compound that carries a terminal polar group [17]. Generally, this phase displays a focal-conic texture.

SmC differs from the SmA phase as the director of the molecule’s constant tilt angle measured normally to the smectic plane [18]. The layers of the smectic C phase are closely packed, respecting the director of the phase, in a short-range. Similar to the nematic mesophase, SmC has a chiral smectic C (SmC*). The director of the layer is identical to SmC, and the only difference it makes is that the angle that rotates from a layer to another and forms a helix. According to [16], there is a sub-phase of the smectic C phase called the anticlinic smectic C. The order of this molecule is similar compared to the order of a smectic C phase. The difference between each phase is the correlation between the tilted direction in the layers.

Understanding of the structure–property relationship of the compound is required in order to understand the interconnections between the core system, linking unit, and terminal group. Currently, there is a lack research focused on this area. There has been a lack of research that focuses on the azo-based compounds, especially on the methods and the structure–property relationship. Hence, this review will help future researchers to understand the phase properties and the characterization of azo-based compounds.

## 2. Structure–Property Relationship

Despite being lyotropic or thermotropic, most liquid crystal can exhibit at least one liquid crystal phase. According to Safinya et al. (1986), a thermotropic liquid crystal can exhibit one to several liquid crystal phases between the crystal and isotropic states [18]. However, there is a time where a liquid crystal does not exhibit any mesophase at all. This study will explain the possibilities of liquid crystal disclosing different types of liquid crystal phase, why a structure with the same central unit gives out different liquid crystal phases, and how any substitution can affect the mesomorphic.

In accordance with a study by Collings and Hird (1997), the structure–properties relationship is crucial to understand for the synthesizing or altering of a molecule in a certain way or arrangement to get a certain mesomorphic phase [14]. This understanding is important, especially when the liquid crystal is synthesized for a particular application that requires a particular liquid crystal phase.

A study was conducted in 1997 by Parra et al. on the structure–properties relationship of azo-based compounds [19]. In this study, Parra and her co-worker synthesized two azo compounds, **1**(a–f) and **2**(a–f). All homologues of compound **1**a were claimed to display a nematic phase. In addition, homologues *n* = 9 and 10, compounds **1**e and **1**f, respectively, show a monotropic smectic C phase. Similar to compound **1**, all homologues of compound **2** also exhibit an enantiotropic nematic phase. However, compounds **2**(a–c) displays a monotropic smectic C while compound **2**(d–f) exhibit an enantiotropic smectic C phase. Compound **2** appears to have more extensive mesomorphic range compared to compound **1**. Parra and her co-worker stated that the low mesophase stability of compound **1** is caused by the presence of a thiophene ring which produces an additional deviation that hinders the formation of a stable mesophase.

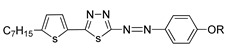
**Comp****1a****1b****1c****1d****1e****1f**RC_5_H_11_C_6_H_13_C_7_H_15_C_8_H_17_C_9_H_19_C_10_H_21_



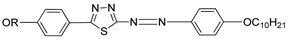


**Comp**

**2a**

**2b**

**2c**

**2d**

**2e**

**2f**
RC_5_H_11_C_6_H_13_C_7_H_15_C_8_H_17_C_9_H_19_C_10_H_21_

Two years later, Belmar et al. (1999) synthesized quite a similar azo compound, namely compounds **3**(a–h) [20]. All homologues of compound **3** give out a nematic phase showing that corresponding molecular interactions occur that give the same outcome in terms of molecular arrangement and thermal stability. However, not all homologues exhibit smectic C phase, such as compounds **3**(a–d) as a layered smectic order is not ideal due to the difference in the volume occupied by the opposite chain. Nevertheless, a tilted smectic C order is formed as the alkoxy chain lengthens starting of homologues 7 to 10, and almost the same to the opposite chain.

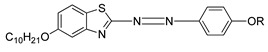
**Comp****3a****3b****3c****3d****3e****3f****3g****3h**RC_3_H_7_C_4_H_9_C_5_H_11_C_6_H_13_C_7_H_15_C_8_H_17_C_9_H_19_C_10_H_21_

In 2001, Lee et al. synthesized two crystalline dyes containing a non-activated pyranylazo group, compounds **4**(a–f) and **5**(a–f) [21]. At the first heating phase, none of the homologues of compounds **4** and **5** display any mesomorphic property due to the thermo- and photochromic properties of the dye. This property causes a small ring-opened merocyanine species, which is not favored at high temperatures. Only after the first heating-cooling phase does the compound starting to exhibit a mesophase. Compounds **4**a and **4**c show an enantiotropic nematic phase, compounds **4**b and **4**d display a monotropic nematic phase, while compounds **4**e and **4**f do not form any liquid crystal phase. In contrast, all homologues of compound **5** form a nematic phase, and some even display a smectic A phase, except compounds **5**d and **5**e.

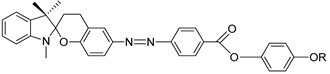
**Comp****4a****4b****4c****4d****4e****4f**R(CH_2_)_4_CH_3_(CH_2_)_5_CH_3_(CH_2_)_6_CH_3_(CH_2_)_7_CH_3_(CH_2_)_8_CH_3_(CH_2_)_9_CH_3_



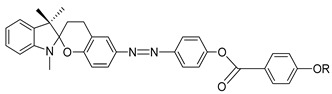


**Comp**

**5a**

**5b**

**5c**

**5d**

**5e**

**5f**
R(CH_2_)_4_CH_3_(CH_2_)_5_CH_3_(CH_2_)_6_CH_3_(CH_2_)_7_CH_3_(CH_2_)_8_CH_3_(CH_2_)_9_CH_3_

In the same year, two azo compounds were synthesized containing a similar structural unit, comprised of pyridine in compounds **6**(a–f) and 1,3,4-thiadiazole rings in both compounds **6**(a–f) and **7**(a–f) [22]. All homologues of both compounds exhibit a liquid crystal phase. As for compound **6**, all homologues display a crystal to isotropic transition, and only two of the highest homologues, compounds **6**e and **6**f exhibit a monotropic nematic phase. Compound **7** shows a dimorphism of a nematic phase and a smectic C phase. However, compounds **7**(a–c) exhibit a monotropic smectic C phase. Nevertheless, the thermal stability of compound **7** is higher compared to the thermal stability of compound **6**. The most prominent difference between compounds **6** and **7** can be seen from the structure. Compound **6** comprises a pyridine unit at the end of the central core and only one lateral chain, while compound **7** has a benzene ring instead of a pyridine, two lateral units, and a greater molecular length. To briefly summarize, compound **6** contains a pyridine unit, and is not long enough to be polarized enough to exhibit a stable liquid crystal phase compared to compound **7**.

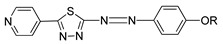
**Comp****6a****6b****6c****6d****6e****6f**RC_5_H_11_C_6_H_13_C_7_H_15_C_8_H_17_C_9_H_19_C_10_H_21_



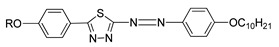


**Comp**

**7a**

**7b**

**7c**

**7d**

**7e**

**7f**
RC_5_H_11_C_6_H_13_C_7_H_15_C_8_H_17_C_9_H_19_C_10_H_21_

So et al. (2003) reported in his study about compounds **8**(a–d), where this compound has two two-ring mesogenic units that are connected by a spacer, namely 2-hydroxy-1,3-dioxypropylene [23]. All homologues of compound **8** have display an enantiotropic schlieren and/or a broken fan textured smectic C phase except for compound **8**a. Figure 1 shows a micrograph of compound **8**c. This phenomenon happens due to the temperature range of mesophase increase with the increasing length of the terminal alkyl chain.

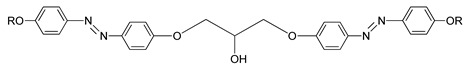
**Comp****8a****8b****8c****8d**RC_6_H_13_C_8_H_17_C_10_H_21_C_12_H_25_

Compounds **9**(a–c) was synthesized by Abbasi et al. (2006) [24]. Compounds **9**(a–b) display a monotropic liquid crystal phase during cooling from the isotropic liquid. However, compound **9**c formed an enantiotropic mesophase behavior and showed a liquid crystalline characteristic on heating and cooling from the isotropic liquid. Abbasi et al. declared that the stability of mesophase is influenced by the alkyl length (R). Compound **9**c with the longest alkyl chain length exhibit enantiotropic mesophase which is thermodynamically stable compound, and compound with a short chain length (**9**a and **9**b) exhibit unstable mesomorphic behavior.

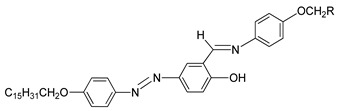
**Comp****9a****9b****9c**RC_7_H_15_C_11_H_23_C_15_H_31_

In the same year, compounds **10**(a–c) and **11**(a–c) were synthesized by Rezvani et al. (2006) [25]. Compounds **10**(a–c) did not show any liquid crystal phase as this compound directly melted into an isotropic liquid. It was claimed that the melting point of the ligands decreases with increasing alkyl chain length value. Compound **11**a exhibit four endothermic transitions, where the first two transitions are equivalent to the crystal-to-crystal transition. The third transition peak at 204.3 °C with enthalpy values of 532.43 kJ mol^−1^ is corresponding to a crystal phase to a mesophase. A Schlieren texture typically for the smectic C phase was detected, as reported in Figure 2. The fourth transition peak at 240.3 °C with a low enthalpy value of 5.25 kJ mol^−1^ is responsible for the transition from mesophase to the isotropic liquid. The high clearing enthalpy corresponding to the mesophase to isotropic transition indicates that the mesophase structure is highly in order.

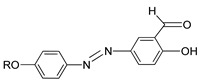
**Comp****10a****10b****10c**RC_9_H_19_C_11_H_23_C_13_H_27_



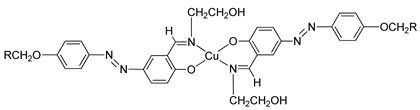


**Comp**

**11a**

**11b**

**11c**
RC_9_H_19_C_11_H_23_C_13_H_27_

Compounds **12**(a–c) were first synthesized by So et al. in 2001. In 2006, he conducted another study using the same compound but with a different alkyl chain [26]. During the heating phase, compound with the highest homologues, compound **12**c, exhibits a crystal to smectic C phase at 163.0 °C, and smectic C to isotropic liquid at 168.0 °C and compounds **12**a and **12**b does not show any mesomorphic phase. However, during the cooling scan, compounds **12**b and **12**c show a smectic C mesophase (Figure 3) but not compound **12**a. Compound **12**b form a monotropic smectic C liquid crystal phase with a schlieren or broken-fan texture, whereas compound **12**c exhibit an enantiotropic smectic C liquid crystal phase with a schlieren texture and compound **12**a shows no mesomorphic behavior. So et al. (2006) reported that increasing the alkyl chain is expected to increase the length-to-breadth, resulting in a liquid crystal phase, especially the smectic phase [27]. He also claimed that as the length of the terminal chain increase, the temperature range of the smectic phase also increases. It was also discussed in the study that the entropy change keeps increasing as the terminal chain increase, hence, a smectic mesophase is prone to show, and as the terminal chain increases, the thermal stability of a tilted smectic C also increases.

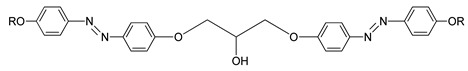
**Comp****12a****12b****12c**RC_5_H_11_C_7_H_15_C_9_H_19_

Lutfor et al. (2009) have synthesized compounds **13**(a–f) with six different homologues [28]. As for raw compound **13**a, two transition peaks were observed during cooling of compound at 175.2 and 123.1 °C, which are assigned for the isotropic to smectic and smectic to crystal transition, respectively. Chloro-substituted compound **13**b also shows two peaks on cooling, 144.3 and 126.0 °C, which are corresponded to isotropic to smectic and smectic to crystal transition, respectively. Compound **13**c and **13**f do not display any liquid crystal phase, and it melts at 168.0 and 165.0 °C and crystallized at 162.0 °C and 160.0 °C, respectively. Compound **13**d is a fluoro-substituted compound that displays two peaks on cooling at exactly 150.5 and 127.1°C, corresponding to isotropic to smectic and smectic to crystal transition, respectively. Compound **13**e is a fluoro, and chloro-substituted compound that shows two peaks on cooling at 142.9 and 126.3 °C, correspond to isotropic to smectic phase and smectic to crystal phase. In summary, the substitution of F atom to the aromatic core in compound **13**d and chloro-substituted of compound **13**b decreases the mesophase-isotropic transition temperature compared to compound **13**a. Compound **13**e possessing both F and Cl atoms have the lowest transition temperature compared to the other compounds.

When cooling from the isotropic liquid, compounds **13**d and **13**e exhibit a fan-like texture which is typical for a smectic phase. Both compounds form a smectic phase at a lower temperature due to the addition of F atoms in compound **13**d and F and Cl atoms in compound **13**e. Compound **13**(a, b, d, and e) were found to be monotropic in nature (Figure 4), and only compound **13a** was studied to be thermodynamically more stable compared to the other compound. However, it was expected that compound **13a** has the highest mesomorphic stability as it contains no lateral substitution that may disturb the aromatic core packing.

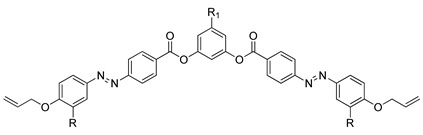
**Comp****13a****13b****13c****13d****13e****13f**RHHHFFFR’HClCOOHHClCOOH

There are several possible reasons for the absence of a smectic mesophase. Firstly, the dipole of the substituent is partly canceled by the dipole of the ester group. Next, no other dipole across the long axis of molecules. Lastly, the strength of intermolecular lateral attractions is reduced as a result of a long narrow to be pushed further apart due to an increase in molecular breadth [29]. Compounds **14**(a–b) consists of CH_3_ and F substituted compounds, respectively. Compound **14**a is prone to show a nematic phase while compound **14**b tend to exhibit both nematic and smectic phase. Briefly summarize, the ratio of lateral to the terminal attraction of compound **14**b is higher than compound **14**a. This is because the F atom does not increase the molecule width due to the small atomic size compared to the adjacent substituent. Although the C–F bond has a high dipole moment, the bond itself cannot be entirely canceled by the dipole moment because of the ester group that presents in the compound. The dipole moment of the long axis will approach the dipole moment of the lateral axis, causing the attractive terminal forces is almost the same as the attractive lateral forces. Hence, two phases were formed at low temperatures.

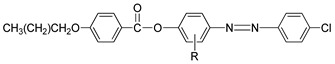
**Comp****14a****14b**RCH_3_F

In 2011, two ligands containing Cu and Ni atoms were synthesized by Yeap et al. (2011), namely compounds **15**(a–e) and **16**(a–e) [30]. All compound **15** first exhibits a schlieren textured nematic phase followed by a focal conic fan shape textured typically for smectic A. However, all compound **16** does not exhibit a mesomorphic behavior due to direct isotropization during the heating and cooling phase without exhibiting any liquid crystal phase. The layer spacing of compound **15**(a–e) is 1.04 Å. This layer spacing corresponds to the molecular length, and the layer thickness of compound **15** is slightly lower than its molecular length. This explains the presence of interdigitation of the alkoxy chain and its neighboring layer.

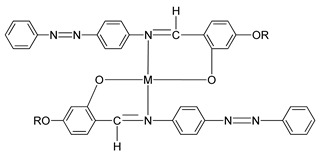
**Comp****15a****15b****15c****15d****15e**MCuCuCuCuCuRC_8_H_17_C_10_H_21_C_12_H_25_C_14_H_29_C_16_H_33_**Comp****16a****16b****16c****16d****16e**MNiNiNiNiNiRC_8_H_17_C_10_H_21_C_12_H_25_C_14_H_29_C_16_H_33_

Yeap et al. (2011) also focused on the study of a structure–property relationship of compound **17**, which was synthesized by Reddy et al. (1991) [30,31]. Compound **17** exhibit a smectic A phase and is thermodynamically enantiotropic up to the clearing temperature of 33.0–261.0 °C. Such a thing happens due to the introduction of azobenzene cored that may induce mesophase formation in compound **17** and lower the temperature of complexes.

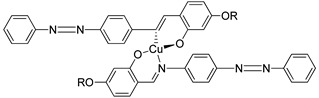
**Comp****17**RC_5_H_11_

A year later, compounds **18**(a–d) were synthesized by Yang et al. (2012) [32]. In this study, all compounds **18** exhibit mesomorphic phases. This is due to the six or eleven segments (*n* = 6 or 11) present in the compound that acts as a spacer. In his study, Yang et al. (2012) argued that in order for a compound to exhibit a mesomorphic phase, a liquid crystal must contain a suitable spacer length between the center and the terminal chain [32]. This flexible spacer is also crucial as it affects the dipole and molecular interaction between the compound, which will play an important role in mesophase formation. During the heating scan, compound **18**a melted, and a cholesteric liquid crystal was exhibited between 138.7 and 189.1 °C. On the other hand, compound **18**c was observed to show an enantiotropic mesophase at between 160.0–185.4 °C and 135.6–183.1 °C, upon heating and cooling scan, respectively. Compound **18**a displayed a broader phase transition temperature range compared to **18**c, which shows that the electron-withdrawing nitro moiety intensifies the head-to-tail molecular interactions. However, in compound **18**b, the longer methylene spacer decreases the head-to-tail interactions, resulting in a phase transition temperature range. In contrast, compounds **18**c and **18**d that contain a methoxy terminal chain exhibit different behavior. Compound **18**d has a broader phase transition temperature change compared to compound **18**c. Based on this observation, the rigidity of the mesomorphic core, length of the flexible spacer, and the type of terminal chain play an important role in the dipole–dipole interaction that leads to the variability of the phase transition temperature. Hence, it will later influence the formation of different types of liquid crystal properties and textures.

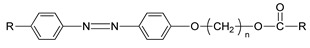
**Comp****18a****18b****18c****18d**n611611RNO_2_NO_2_OCH_3_OCH_3_

Two azo compounds bearing two different moieties used at the end terminal were synthesized: compounds **19**(a–i) and **20**(a–i). Lower homologues of both compounds only exhibit a nematic phase, while compounds with higher homologues, which are compounds **19**h, **19**i, **20**f, **20**g, **20**h, and **20**i, possess a longer terminal chain exhibit an additional phase transition directing to a layered mesophase. During the cooling scan, all homologues of both compounds are enantiotropic in nature and exhibit a schlieren and marbled texture typically for a nematic mesophase. As observed, the nematic phase range decreases with an increasing chain length in both compounds [33].

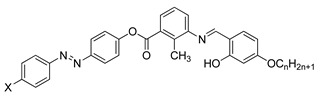
**Comp****19a****19b****19c****19d****19e****19f****19g****19h****19i**n456789101112XCNCNCNCNCNCNCNCNCN**Comp****20a****20b****20c****20d****20e****20f****20g****20h****20i**n456789101112XNO_2_NO_2_NO_2_NO_2_NO_2_NO_2_NO_2_NO_2_NO_2_

Compounds **21**(a–d) have a similar molecular structure except for the R group that was substituted on the benzothiazole ring. Hence, different types of mesophase were displayed by each compound during the heating and cooling cycles. Generally, the terminal methacrylate unit and the substituent on the benzothiazole ring are crucial to forming a liquid crystal. Compound **21**a shows only a smectic mesophase, while compounds **21**b, **21**c, and **21**d display both nematic and smectic phases. The terminal methacrylate unit is responsible for forming the smectic phase as this unit elevates the polarizability anisotropy, which is ideal for the lateral attraction of molecules to form a strong smectic phase. As mentioned earlier, compounds **21**b, **21**c, and **21**d exhibited both smectic and nematic liquid crystal phases. Nematic liquid crystal is favored as the replacement of hydrogen atom on compound **21**a by a methyl group on compound **21**b, methoxy group on compound **21**c, and ethoxy group on compound **21**d, to the sixth position of the benzothiazole ring was facilitated by conjugation of the high core polarizability with the short terminal chain [34]. The polarization of benzothiazole moiety is directly affected by the electron distribution in the electron-donating substituent. The substituent size also influences the mesophase temperature transition in the sixth position of the benzothiazole ring. The ethoxy substituent has the greatest mesophase stability compared to the methoxy or methyl substituent. To conclude, the terminal methacrylate unit and the sixth position substitution play a vital role in confirming the liquid crystal phase formation.

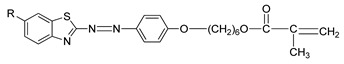
**Comp****21a****21b****21c****21d**RHCH3OCH3OC2H5

Compounds **22** and **23** are azobenzene chromophores with a fluoro substituent. Both compounds are similar compared to the number of fluorine atom each compound possess, compound **22** contain only one fluorine atom while compound **23** contain two fluorine atoms. Compound **22** shows a nematic to isotropic transition at 158.9 °C, whereas compound **23** exhibit a smectic A to isotropic transition at exactly 157.6 °C. The number of fluorine substituent present in the compound influence the transition temperature by increasing the number of fluorine atom, the transition temperature will decrease [35].

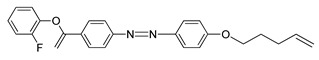
**Compound 22**



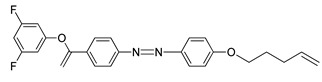


**Compound 23**


All compounds **24**(a–f) exhibited a schlieren texture in an enantiotropic liquid crystal phase which is typical for a nematic phase. The thermal transition of the nematic phase of methoxy homologues (**24**d, **24**e, and **24**f) is higher than the methyl homologues compounds (**24**a, **24**b, and **24**c). The presence of an oxygen atom on the methoxy group is vital domination in the mesophase thermal stability and the thermal phase range. Upon heating, the enthalpy values of the crystal to the nematic phase transition of compounds **24**a, **24**b, and **24**c are 38.8, 46.6, and 50.3 °C, respectively. As for compounds **24**d, **24**e, and **24**f are 99.0, 77.8, and 72.4 °C, respectively. These large enthalpy values suggest a strong Van der Waals interaction between the end unit of an adjacent molecule. However, upon cooling, the enthalpy values recorded for the crystal to the nematic phase transition are 26.2, 27.1, 25.8, 46.6, 43.7, and 47.0 °C for each compound, respectively. These lower enthalpy values upon cooling are lower compared to the enthalpy values upon heating. This is due to the fact that the strong Van der Waals interactions between the end of the chain of a neighboring molecule keep them static in the crystalline state before the gradation to liquid crystal starts in the heating cycle [36].

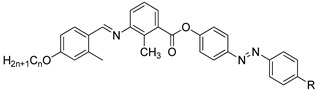
**Comp****24a****24b****24c****24d****24e****24f**n456456RCH3CH3CH3OCH3OCH3OCH3

Compounds **25**(a,b) are novel benzoates that contain three phenyl rings and a trimethylsilyl group in the terminal position. Both compounds possess a similar structural formula having a central azo compound with an eminently polar nitro group and a pentyl chain except the R group at the end of the chain. Despite having a similar structural formula, only one compound exhibits a liquid crystal property, which is compound **25**a. The optical texture of compound **25**a is illustrated in Figure 5. The polar nitro terminal favored a smectic A phase, while compound **25**b possessing an alkyl chain does not display any liquid crystalline phase property. To conclude, the property of a liquid crystal is reduced as the chain length lengthen as per the disruption of the packing layers. Hence, the only molecule with a short-chain compound can display any liquid crystalline property [37].

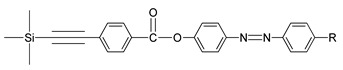
**Comp****25a****25b**RNO_2_C_5_H_11_

Compounds **26**(a–e) were synthesized by Selvarasu and Kannan, in 2015 [38]. Generally, the transition temperature of compounds **26**a and **26**b is larger compared to the transition temperature range of compounds **26**d and **26**e, as the electron-withdrawing substituent in compounds **26**a and **26**b have a strong Π to Π* interaction that increase the head to tail molecular structure. The nematic to the isotropic thermal stability of compounds **26**a and **26**b are higher compared to compounds **26**d and **26**e. The reason is that the –CN and –Cl groups as well as the benzene ring located in the terminal position will cause both compounds to have high polarity and thermal attraction that later will causes the thermal stability to increase. The only difference that compounds **26**a and **26**e show is on the substitution unit at the terminal end. Both of these compounds share the same central unit, a cinnamoyloxy group (–C_6_H_5_–CH=CH–COO–) which is directly connected to the alkyloxyphenylester group. The presence of the double bond in the cinnamoyloxy group lengthen the length of polarizability and elevate the thermal stability.

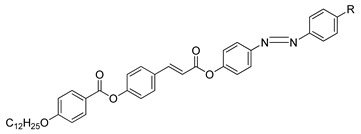
**Comp****26a****26b****26c****26d****26e**RCNClHCH_3_OCH_3_

A year later, Selvarasu and Kannan (2016) synthesized a new azobenzene compound, compound **27** [39]. Compounds **27**a, **27**b, and **27**c exhibit a nematic phase with the lowest homologues of n: 6 forming small droplets that affiliate into a classic schlieren nematic and compound of homologues n: 8 and 10 display a focal-conic textured nematic phase while the compound with the highest homologues exhibits a nematic phase and an addition of a focal conic textured smectic C phase upon cooling from the isotropic liquid. The formation of a nematic phase is a result of the alkoxy chain terminal being more noticeable. Compound **27** contains a cinnamoyloxy group connected directly to the azo compound and the ester group that acts as the central unit of the molecule, and a double bond that increase the polarizability. The core of compound **27** is directly bonded to an ester group, causing the molecule to lose stability. This phenomenon is due to the ability of the oxygen atom of the carbonyl group to bump into the non-bonded sides of the adjacent hydrogen in the aromatic ring, which attributed to the strain in the molecules. An ester linkage contains a ketonic bond, where the electron is being pulled by the ester group which causes the transition temperature to decline. To sum everything up, the attributes responsible for mesophase formation are the effect of terminal alkoxy containing azobenzene moiety and the central linkage with or without the spacer. The presence of an olefinic unit reinforces the length of polarizability, and an additional olefinic unit would proliferate the thermal stability with a high polarizability.

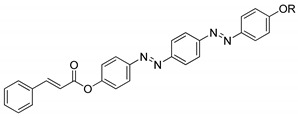
**Comp****27a****27b****27c****27d**RC_6_H_13_C_8_H_17_C_10_H21C_12_H_25_

Compounds **28**(a–f) are azo-bridged compounds with triphenylene cores. The only difference between the compounds **28**(**a**–**f**) is the length of the methylene units. The length of the spacer has a vital role in the fluctuation of the transition temperature. Upon cooling, the melting point of compound **28**a is higher compared to compound **28**b. However, it gradually elevates from compounds **28**b to **28**e. According to Yeap et al. (2016), it has something to do with the spacer length [40]. As the length of spacer increase, the melting point continue to decrease from compounds **28**e to **28**f. Nonetheless, the clearing temperature of compound increase from compounds **28**b to **28**c and keep on descending as the length of alkyl chain increase. By reason, it may be due to the dilution of the mesogenic core, which causes the spacer to be flexible.

Compound **28**f exhibits a smectic C phase, where it has been claimed that the formation of a smectic phase results from a side-by-side organization of azobenzene base peripheral units that are adjacent in parallel to each other. Many flexible spacers link all homologues of compound **28**. This flexible spacer enables a conformational change so that a rod-shaped unit could align in a parallel arrangement resulting in a layered structure of a smectic phase [40].

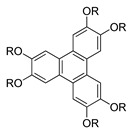
**Comp****28a****28b****28c****28d****28e****28f**n5678910R
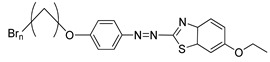


Compounds **29**(a–c) were synthesized by Sen et al. (2016) [41]. All of compounds **29**(a–c) exhibit only one mesophase transition (Figure 6). Uncommonly for a rod-shaped molecule, the enthalpy values for the liquid crystalline phase of the isotropic liquid are higher than the enthalpy values of a solid–liquid crystalline phase. This unusual phenomenon may be due to the molecule’s arrangement within each layer, which the minimized steric forces have driven.

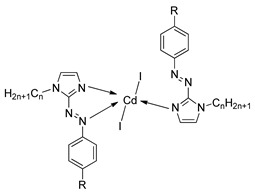
**Comp****29a****29b****29c**n101218RHHH

Jaworska et al. (2017) synthesized compounds **30**(a–n) [42]. Out of 14 compounds with different homologues, he showed that two of them do not exhibit any liquid crystalline property. Compounds **30**a and **30**b do not possess any liquid crystal phase; by reason both compounds have high melting points compared to other homologues. Other compounds studied exhibit at least one enantiotropic mesophase, which was identified as a focal-conic textured smectic A. Compound **30**c is the first homologue displaying smectic A and the width is 30.0 °C, as it can be seen in Figure 7. Starting from compound **30**d until compound **30**k, the width of the smectic A phase is broader, ranging from 40.0 to 75.0 °C. The smectic range keeps narrowing corresponding to increasing the alkoxy chain to n: 14 and above.

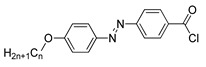
**Comp****30a****30b****30c****30d****30e****30f****30g**n45678910**Comp****30h****30i****30j****30k****30l****30m****30n**n45678910

All compounds **31**(a–e) possess at least one liquid crystal phase except compound **31**e, where it exhibits an extra liquid crystalline phase familiar as a smectic phase. This is a result of the length of the alkoxy chain in compound **31**e is long enough to stabilize the lamellar molecular arrangement ideal for the molecule to display a smectic phase. The variation of the terminal group by the methylene spacer located in 1,4-disubstituted triazole derivatives, the molecular alignment of mesophase is stabilized [43]. Imine as a central unit causes the polarity to increase through retaining of linear patter. Minor changes in the alkyl chain may cause major changes in the transition temperature and types of mesophase formed.

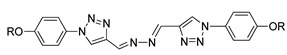
**Comp****31a****31b****31c****31d****31e**RC_6_H_13_C_7_H_15_C_8_H_17_C_9_H_19_C_10_H_21_

In the same year, two identical compounds, **32** and **33**, were synthesized. The only difference between these two compounds is the central unit, where it exchanges place with the adjacent group. The melting point and the clearing point of compounds **32** and **33** are 104.8, 130.2, and 166.5, 162.2 °C, respectively. These temperature changes may be due to the length to diameter ratio, which eventually increases the transition temperature [44].

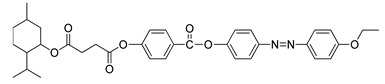
**Compound 32**



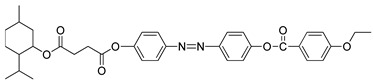


**Compound 33**


Each homologue of compounds **34**(a–c) exhibit a different liquid crystal phase despite having similar compound. The formation of different mesophase within this compound is influenced by the substitution terminal (–R). Compound **33**a contains an end non-polar methyl group, and it shows a nematic phase at 31.7 °C upon cooling. Compound **33**b comprises a relatively polar chloro moiety, revealing a nematic phase about 3.0 °C above the isotropic liquid and a smectic A phase at 47.0 °C upon cooling. A polar nitro moiety is present in compound **33**c, a long-range smectic A phase is observed at 106.3 °C in the cooling scan. The polarity of substituent not only affects the formation of different mesophase, but also caused the clearing temperature to change. The mesophase range increase as the polarity of substituent increase from **33**a to **33**c [45].

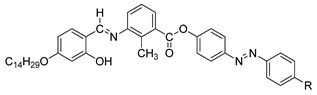
**Comp****34a****34b****34c**RCH_3_ClNO_2_

Compounds **35**(a–e) were synthesized in 2019 by Madiahlagan et al., and are a series of azo-coumarin compounds with different aliphatic chain lengths [46]. Lower homologues of compound **35** displayed a nematic phase at 120.0 °C for compound **35**b, while compound **35**e exhibited a broken fan-shaped smectic A at 170.0 °C upon cooling from the isotropic liquid. The difference in mesomorphic formation and the transition temperature is greatly influenced by the increasing carbon atom of the aliphatic substituent.

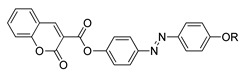
**Comp****35a****35b****35c****35d****35e**RC_6_H_13_C_8_H_17_C_10_H_21_C_12_H_25_C_14_H_29_

The shortest terminal chain, compounds **36**a and **36**e, exhibited three liquid crystal phases which are nematic phase, smectic A phase, and smectic C phase, while compound **36c** containing a butyl terminal chain exhibit a twist grain boundary smectic A (TGBA) rather than a smectic A itself and a smectic C phase. However, the broadest smectic S phase was observed to be of compound **36**b. The TGBA phase does not present in **36**b, and **36**d is due to the odd-even effect. Compound **36**i with the longest aliphatic group displayed a smectic A to smectic C mesophase transition with the temperature narrowed. It was affected by the elongation of the non-chiral alkyl chain, which eventually decreases the mesophase range.

As for compounds **37**(a–i), the TGBA phase appeared in compound **37**a and disappeared from compounds **37**b to **37**d and re-appear in compound **37e** implying that a change has happened for compounds with the middle length alkyl chain. Further studies were done to identify the cause, however, further increase of the carboxylate group leads to a reduction of mesophase property [47].

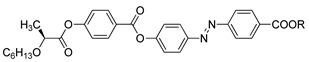
**Comp****36a****36b****36c****36d****36e****36f****36g****36h****36i**RC_2_H_5_C_3_H_7_C_4_H_9_C_5_H_11_C_6_H_13_C_7_H15C_8_H_17_C_10_H_21_C_12_H_25_



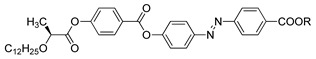


**Comp**

**37a**

**37b**

**37c**

**37d**

**37e**

**37f**

**37g**

**37h**

**37i**
RC_2_H_5_C_3_H_7_C_4_H_9_C_5_H_11_C_6_H_13_C_7_H_15_C_8_H_17_C_10_H_21_C_12_H_25_

In the following year, compounds **38**(a–e) and **39**(a–e) were synthesized, where all compounds studied exhibit a liquid crystalline property. Within compound **38** and all higher homologues were observed to be enantiotropic except compounds **38**a, and **38**b, while compounds **39**a and **39**d are enantiotropic in nature. Compounds **38**b and **38**c display an enantiotropic smectic B1 phase, while compound **38**d exhibits an enantiotropic smectic B1 phase and an addition of a monotropic smectic B2 phase. Theoretically, a change in the alkyl chain will cause the mesophase property to be varied, and the formation of smectic B proves the point [48].

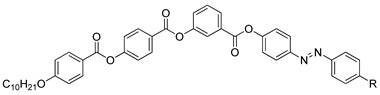
**Comp****38a****38b****38c****38d****38e**RC4H9C6H13C8H17C10H21C12H25**Comp****39a****39b****39c****39d****39e**ROC4H9OC6H13OC8H17OC10H21OC12H25

Sardon et al. (2021) synthesized compounds **40**(a–d) and compared them with a compound studied back in 2016 by Karim et al., labeled as compounds **41**(b–c) [49,50]. The result of this observation is illustrated in Table 1.

Based on Table 1, it can be observed that the mesophase stability is inconsistent. This inconsistency is caused by the substituent (–H, –Cl, –Br, –CN). However, the nematic mesophase range of compounds **40**(b–d) are wider. The substitution of hydrogen atom on the para-position of benzene ring affects the polarizability of mesogens that aid the formation of the liquid crystal phase. Compound **40**d presented the highest nematic to isotropic clearing transition temperature because of higher polarization effects caused by the cyano substitution that gives high temperature to break the association within the molecule.

Compounds **40**b and **40**c only reveal the formation of a nematic phase, while compounds **41**b and **41**c displayed both nematic and smectic A phases. The methyl side chain substituent in compounds **40**b and **40**c truncate the melting and clearing temperature compared to compounds **41**b and **41**c that possess no methyl side chain. Generally, any lateral group attached to the core unit of a compound may cause some changes in the mesomorphic behavior. The presence of methyl side-chain group widens the core unit where the intermolecular separation is eventually increased and leads to lower mesophase stability.

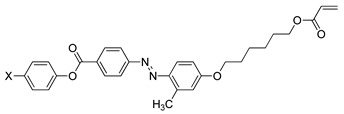
**Comp****40a****40b****40c****40d**RHClBrCN



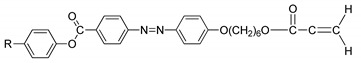


**Comp**

**41a**

**41b**
RClBr

## 3. Application

In this era, everyone uses at least one liquid crystal in their daily life. Over the years, the use of electronic devices has increased to meet the demand of easing our lives. The most common liquid crystal used is the liquid crystal display (LCD) that is present on our mobile phones, computers, laptops, and many more [51,52]. LCD is an electronic display device with a flat panel that uses a liquid crystal as its main operation medium [53]. An LCD possess both nematic and smectic phase. However, a smectic LCD is more demanded compared to the nematic LCD as the response is faster. The display on an LCD is not emitted by the liquid crystal, instead of liquid crystal acts as a reflector that produces the image in either monochrome or in color [27].

Another application is the liquid crystal thermometer. As informed, a thermometer detects the temperature of something. Liquid crystal is applied to this system so that when it detects any temperature, a specific color persists. Recalling from a study by Kitzerow and Bahr (2001), a chiral nematic liquid crystal reflects light of a wavelength of the same length of the pitch [16]. As the pitch is temperature-dependent, the color that is reflected is also temperature-dependent. This set-up is applied to various applications, such as the mood ring that has recently gone viral, where the ring will conjure different color according to our body temperature. In addition to that, this application is ideal for use in the medical area where it can detect different temperature distributions on a tumor patient as the tumor, and its surrounding tissues develop different temperatures [14].

## 4. Conclusions

There are several factors observed that might influence the formation of a different liquid crystal phase. The most common factor that can be scrutinized in most studies conducted by most authors is the increase in the alkyl terminal chain. These factors may have several consequences, which are the main influences on mesophase development. The effects of the increase in the terminal alkyl chain are that it regulates the increase in molecular length-to-breadth, amplifies the entropy chain, enhances thermal stability, and diminishes the clearing temperature. Moreover, the azo linking unit increase the linearity of the molecule and this characteristic able to induce the mesophase transition.

## Figures and Tables

**Figure 1 polymers-13-03462-f001:**
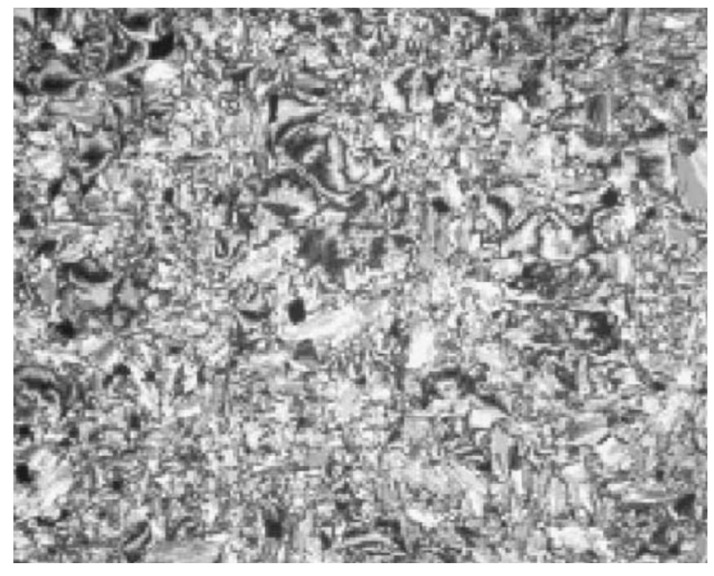
Polarizing optical micrographs upon cooling of isotropic liquid of compound **8**c at 168.0 °C [23].

**Figure 2 polymers-13-03462-f002:**
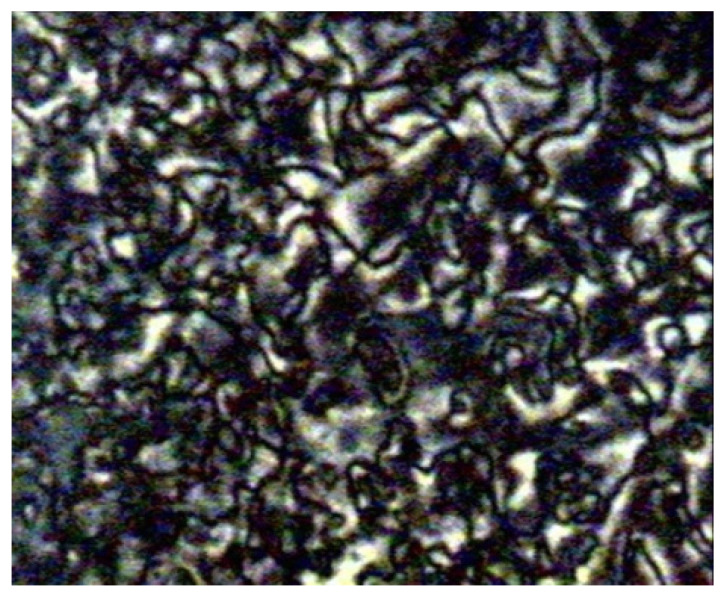
Optical texture of compound **11**a at 210.0 °C [25].

**Figure 3 polymers-13-03462-f003:**
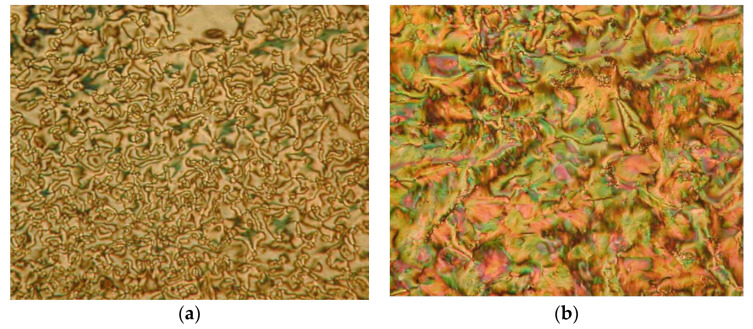
(**a**) Schlieren textured smectic C of compound **12**b at 160.0 °C, and (**b**) Schlieren textured smectic C of compound **12**c at 160.2 °C [26].

**Figure 4 polymers-13-03462-f004:**
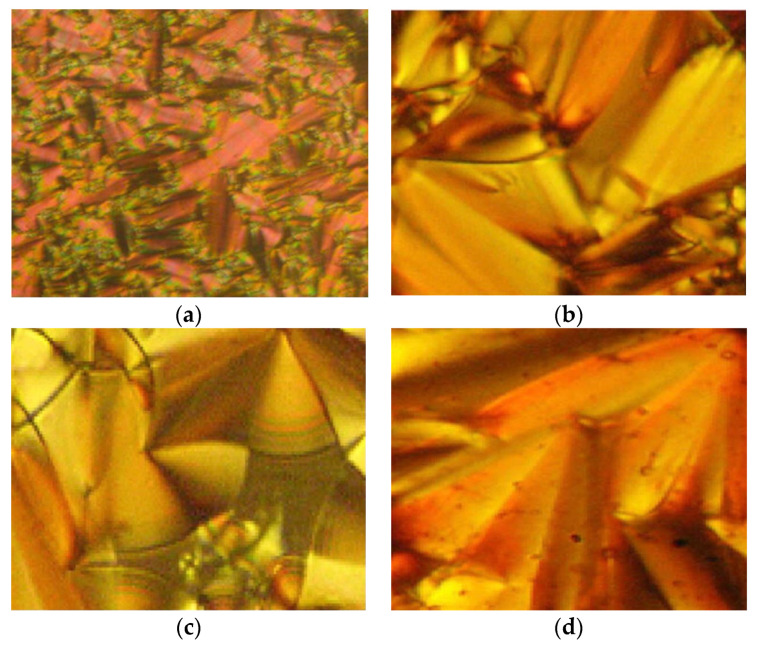
Optical micrograph of (**a**) compound **13**a at 164.0 °C, (**b**) compound **13**b at 130.0 °C, (**c**) compound **13**d at 129.0 °C, and (**d**) compound **13**e at 134.0 °C [28].

**Figure 5 polymers-13-03462-f005:**
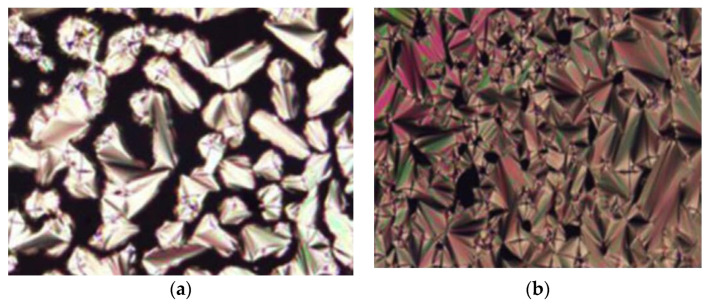
(**a**) Early-stage smectic A compound **25**a at 185 °C, and (**b**) fully grown smectic A phase of compound **25**b at 100.0 °C [37].

**Figure 6 polymers-13-03462-f006:**
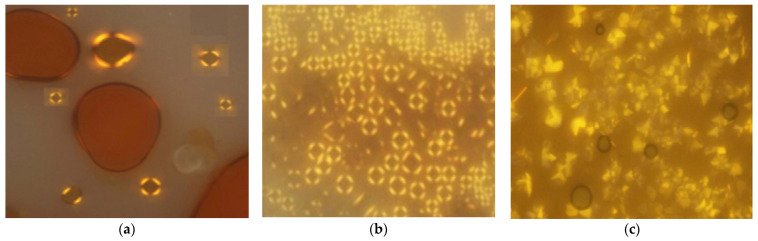
Polarizing micrograph ring-like texture of (**a**) compound **29**a at 123.0 °C, (**b**) compound **29**b at 191.0 °C, and (**c**) cone-like texture of hexatic phase of compound **29**c at 123.0 °C [41].

**Figure 7 polymers-13-03462-f007:**
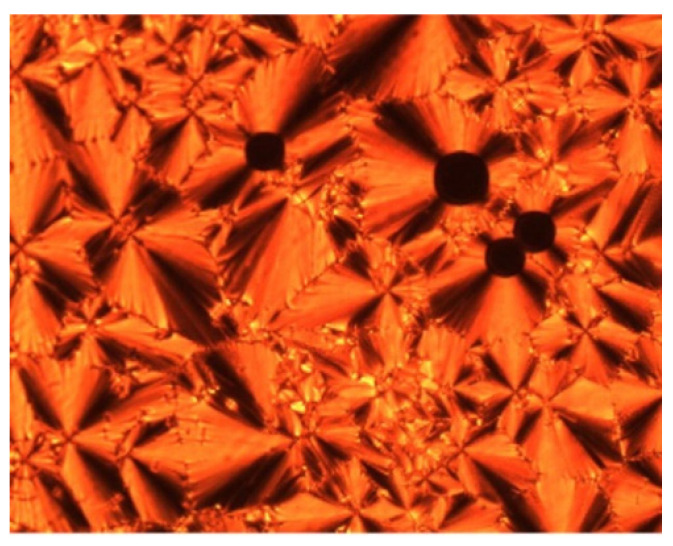
Focal-conic textured smectic A liquid crystalline phase present in compound **30**c [42].

**Table 1 polymers-13-03462-t001:** Phase transition temperature and enthalpy changes of **40**(a–d) and **41**(b–c).

Compound	Scan	Transition Temperature (°C) (Enthalpy Changes (kJ mol^−1^))	Mesophase Range (°C)	
			SmA	N
**40**a	Heat	Cr_1_ 61 (4.5) Cr_2_ 88 (16.1) N 98 (3.7)	-	10
	Cool	I 57 (−0.4) N 18 (−3.8) Cr		
**40**b	Heat	Cr 108 (34.6) N 140 (0.5) I	-	32
	Cool	I 138 (−0.3) N 73 (−33.3) Cr		
**40**c	Heat	Cr_1_ 93 (0.9) Cr_2_ 115 (31.8) N 136 (0.5) I	-	21
	Cool	I 134 (−0.4) N 33 (−15.7) Cr		
**40**d	Heat	Cr 127 (58.4) N 159 (0.9) I	-	32
	Cool	I 156 (−1.2) N 74 (−49.3) Cr		
**41**b	Heat	Cr 116 (33.1) SmA 185 (0.9) N 203 (0.3) I	69	18
	Cool	I 197 (−0.3) N 173 (−0.5) SmA 101 (−15.7)		
**41**c	Heat	Cr 124 (15.7) SmA 192 (1.1) N 204 (0.3) I	68	12
	Cool	I 192 (−0.2) N 174 (−0.3) SmA 112 (−13.0)		

Abbreviations: Cr = crystalline phase; SmA = smectic A phase; N = nematic phase; I = isotropic phase.

## Data Availability

Not applicable.

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
