# Peer review of "Liquid Crystals Investigation Behavior on Azo-Based Compounds: A Review"

_polymers, 2021, doi:10.3390/polym13203462_

Round 1

Reviewer 1 Report

The review entitled "Synthesis, Characterization and Phase Property Investigation of Azo-Based Compounds: A Review" by Nurul Asma Razali and Zuhair Jamain discused factors (the length of the alkyl terminal chain, inter/intra-molecular interaction, presence of spacer, spacer length, polarization effects, odd-even effects, and the presence of an electron-withdrawing group or an electron-donating group) that might influence the formation of a different liquid crystal phase.

The review is well written and interesting, very much. It can be published as is.

Author Response

Dear Reviewer,

Thank you for your recommendation.

Best regards,

Zuhair

Reviewer 2 Report

Azo compounds primary used as dyes for textiles etc. Nowadays, that class of compounds have been intensively investigated regarding their liquid crystalline properties.

The review article of Jamain and Razali provides an impressive survey of the versatile structures of azo compounds possessing liquid crystalline behavior. Structure-property relationships were expounded as well. The subject of the article is interesting, but the manuscript needs some alterations before publishing. First of all, the title “Synthesis, Characterization and Phase Property Investigation of Azo-Based Compounds: A Review” seems to me not appropriate and not concise enough due to following reasons: The article does not give details how the azo compounds were synthesized (reaction equations, synthesis conditions, yields ...). In addition, the term “Phase Property” is much too general and imprecise.

From my point of view section 2 “Materials and Methods“ is dispensable, because it only provides some incoherent details. I disbelieve that this section including table 1 give valuable information to chemists working on synthesis of novel azo compounds. Therefore, section 2 should be rewritten and replenished or it has to be removed from the manuscript at all.

Section 3 supplies lot of molecular structures of azo compounds synthesized during last decades along with the liquid crystalline properties of these compounds and the dependence on structural patterns. Principally, this section is well written but still contains some errors, which have to be corrected.

The following sentence in Section 3 does not make sense (page 6, lines 186,187):Compound with the longest alkyl chain length exhibit a thermodynamically stable compound”,...

Page 7:  The picture of the molecular structure of compound 11 contains an error (one of the hydroxyethyl groups is turned).

Page 11: Structure 18 contains two substituents R. the first one is a nitro group, but the second one is not defined.

Page 11: The position of the substituent R in compound 20 is not defined.

The following sentence at page 13 of section 3 (lines 337,338) does not make sense: “Compounds 22 and 23 were azobenzene chromophores with a fluorobenzene substituent”.

The following sentence at Page 13, line 388 contains an error: “a cinnamoyloxy group (-CH=CH-COO-) which is directly connected to the ...” In fact, the cinnamoyloxy group possesses following formula: C6H5-CH=CH-COO-

Page 13, line 403: the information of following sentence is diffuse, because it is not clear what kind of stability is meant: “The core of compound 27 is directly bonded to an ester group, causing the molecule to lose stability”.

Please note that the triazole derivatives 32 are not really belong to the class of azo compounds.

The quality of the English language of the whole does not meet the standard necessary for a scientific article. There is couple of inappropriate terms and grammar errors. I would therefore recommend that the authors should look for the support of a native English speaker.

Author Response

Response to Reviewer Comments:

Point 1: The review article of Jamain and Razali provides an impressive survey of the versatile structures of azo compounds possessing liquid crystalline behavior. Structure-property relationships were expounded as well. The subject of the article is interesting, but the manuscript needs some alterations before publishing. First of all, the title “Synthesis, Characterization and Phase Property Investigation of Azo-Based Compounds: A Review” seems to me not appropriate and not concise enough due to following reasons: The article does not give details how the azo compounds were synthesized (reaction equations, synthesis conditions, yields ...). In addition, the term “Phase Property” is much too general and imprecise.

Response 1: The title of the article has been revised, followed the content of the manuscript (review paper)

Point 2: From my point of view section 2 “Materials and Methods“ is dispensable, because it only provides some incoherent details. I disbelieve that this section including table 1 give valuable information to chemists working on synthesis of novel azo compounds. Therefore, section 2 should be rewritten and replenished or it has to be removed from the manuscript at all.

Response 2: Section 2 has been removed.

Point 3: The following sentence in Section 3 does not make sense (page 6, lines 186,187):Compound with the longest alkyl chain length exhibit a thermodynamically stable compound”,...

Response 3: The statement has been revised: Compound 9c with the longest alkyl chain length exhibit enantiotropic mesophase which is thermodynamically stable compound, and compound with a short chain length (9a and 9b) exhibit unstable mesomorphic behaviour (line 178)

Point 4: Page 7:  The picture of the molecular structure of compound 11 contains an error (one of the hydroxyethyl groups is turned).

Response 4: The picture for compound 11 has been corrected (line 196)

Point 5: Page 11: Structure 18 contains two substituents R. the first one is a nitro group, but the second one is not defined.

Response 5: The R group on the structure have been revised (line 297)

Point 6: Page 11: The position of the substituent R in compound 20 is not defined.

Response 6: The position of the substituent R in compound 20 has been replaced with X (line 307)

Point 7: The following sentence at page 13 of section 3 (lines 337,338) does not make sense: “Compounds 22 and 23 were azobenzene chromophores with a fluorobenzene substituent”.

Response 7: The statement has been revised: Compounds 22 and 23 are azobenzene chromophores with a fluoro substituent (line 330)

Point 8: The following sentence at Page 13, line 388 contains an error: “a cinnamoyloxy group (-CH=CH-COO-) which is directly connected to the ...” In fact, the cinnamoyloxy group possesses following formula: C6H5-CH=CH-COO-

Response 8: The formula has been replaced with -C6H5-CH=CH-COO- (line 380)

Point 9: Page 13, line 403: the information of following sentence is diffuse, because it is not clear what kind of stability is meant: “The core of compound 27 is directly bonded to an ester group, causing the molecule to lose stability”.

Response 8: Additional information has been added: This phenomenon due to the ability of the oxygen atom of the carbonyl group to bump into the non-bonded sides of the adjacent hydrogen in the aromatic ring, which attributed to the strain in the molecules (line 395)

Point 10: Please note that the triazole derivatives 32 are not really belong to the class of azo compounds.

Response 10: Compound 32 is an azo derivatives (N=N)

Point 11: The quality of the English language of the whole does not meet the standard necessary for a scientific article. There is couple of inappropriate terms and grammar errors. I would therefore recommend that the authors should look for the support of a native English speaker.

Response 11: The grammatical errors in the article have been revised.

Round 2

Reviewer 2 Report

The authors followed most of the suggestions of the reviewer. First of all, the new title matches better to the content of the manuscript. Secondly, section “Materials and Methods“ has been removed from the manuscript. In addition, some errors were corrected.

Unfortunately, the authors did not polish up the quality of the English language. Nevertheless, I recommend the review article for publishing.